# Using N-doped Carbon Dots Prepared Rapidly by Microwave Digestion as Nanoprobes and Nanocatalysts for Fluorescence Determination of Ultratrace Isocarbophos with Label-Free Aptamers

**DOI:** 10.3390/nano9020223

**Published:** 2019-02-07

**Authors:** Xin Li, Xin Jiang, Qingye Liu, Aihui Liang, Zhiliang Jiang

**Affiliations:** 1Key Laboratory of Ecology of Rare and Endangered Species and Environmental Protection (Guangxi Normal University), Guilin 541004, China; xinlistar@163.com (X.L.); kirason0217@126.com (X.J.); ahliang2008@163.com (A.L.); 2Ministry of Education, Guangxi Key Laboratory of Environmental Pollution Control Theory and Technology, Guilin 541004, China

**Keywords:** carbon dot catalysis, TMB, fluorescence, aptamer, isocarbophos

## Abstract

The strongly fluorescent and highly catalytic N-doped carbon dots (CD_N_) were rapidly prepared by a microwave irradiation procedure and were characterized by electron microscopy (EM), laser scattering, infrared spectroscopy (IR), and by their fluorescence spectrum. It was found that the CD_N_ had a strong catalytic effect on the fluorescence reaction of 3,3′,5,5′-tetramethylbenzidine hydroxide ((TMB)–H_2_O_2_) which produced the oxidation product of TMB (TMB_OX_) with strong fluorescence at 406 nm. The aptamer (Apt) was adsorbed on the CD_N_ surfaces which weakened the fluorescence intensity due to the inhibition of catalytic activity. When the target molecule isocarbophos (IPS) was added, it reacted with the Apt to form a stable conjugate and free CD_N_ which restored the catalytic activity to enhance the fluorescence. Using TMB_OX_ as a fluorescent probe, a highly sensitive nanocatalytic method for determination of 0.025–1.5 μg/L IPS was established with a detection limit of 0.015 μg/L. Coupling the CD_N_ fluorescent probe with the Apt–IPS reaction, a new CD fluorescence method was established for the simple and rapid determination of 0.25–1.5 μg/L IPS with a detection limit of 0.11 μg/L.

## 1. Introduction

Nucleic acid aptamers (Apt) can specifically bind to target molecules and have been applied in genomics, food safety, medical diagnosis, biomedicine, and biological analysis [1]. Using Apt-modified metal nanoparticles, analyses such as sensitive Apt nanophotometry, fluorescence methods, resonance Rayleigh scattering, and surface enhanced Raman scattering (SERS) were conducted [1,2,3]. Du et al. [4] prepared a gonadotropin progesterone (P4) Apt-gold nanoparticle colorimetric sensor with a detection range of 2.6–1400 nmol/L P4 and a detection limit of 2.6 nmol/L. Ma et al. [5] obtained a stable tobramycin Apt-nanogold resonance Rayleigh scattering (RRS) probe by using tobramycin (TBC) Apt-modified nanogold with a detect range of 1.9–58.3 ng/mL TBC and a detection limit of 0.8 ng/mL. Deng et al. [6] used specific functionalized Apt complexes on human liver cancer cells, by means of real-time SERS and dark field imaging technology based on gold nanorod targeting probes. Li et al. [7] prepared an Apt-silver conjugate imaging agent (Ag–Sgc8–FAM) with fluorescence. Metal nanoparticles, as we know, have strong surface plasmon effects, but poor stability. Recently, nonmetal nanoparticles such as graphene and silicon were coupled with Apt fluorescence analysis. Graphene nanoparticles (GN) are spherical and laminated. They are the ideal fluorescent nanoquenchers for fluorophores. A new Apt sensor based on fluorescence resonance energy transfer has been developed to detect 2–800 ng/mL 17β-estradiol (E2) by using GN as a fluorescent nanoquencher and shorter E2-specific Apt as a sensing probe with a detection limit of 1.02 ng/mL [8]. Xiao et al. [9] came up with a fluorescence sensing method for 30–900 pg/mL AFB1 with a detection limit of 8 pg/mL by preparing a hairpin structure of a G–quadruplex–Apt chimera which was coated with streptavidin and N-methyl porphyrin IX (NMM) silica nanoparticles. Dehghani et al. [10] designed a fluorescent Apt sensor for the detection of 24.75–137.15 nM kanamycin with a detection limit of 7.5 nM by using somatic/complementary strand- (dsDNA) capped mesoporous silica nanoparticles (MSNs) and rhodamine B as fluorescent probes. Labeling Apt with organic fluorescent molecules rather than nano-labeling has also been reported. A fluorescein-labeled Apt sensor for detecting β-lactamase in milk was constructed with a detection range of 1–46 U/mL and a detection limit of 0.5 U/mL [11]. However, using fluorescent molecules to label Apt has its disadvantages which resulted in reduced selectivity of the Apt reaction and complicated labeling processes. As a new type of fluorescent nanomaterial, carbon dots (CDs) have received great attention due to their excellent optical properties, good chemical stability, low toxicity, excellent biocompatibility, and surface function adjustability. It has become the most popular carbon nanomaterial after fullerene, carbon nanotubes, and graphene, and has been used in bioimaging, fluorescence sensors, energy conversion, environmental monitoring, and nanomaterials [12,13,14]. The research on the preparation of carbon dots has always been one of the research hotspots in nanomaterials and analytical chemistry. A series of carbon dot synthesis methods have been established, such as arc discharge [15], laser etching [16], chemical oxidation [17,18], template [19], microwave [20,21,22], and hydrothermal procedures [23], and the microwave method has attracted much attention due to its rapid preparation speed and use of harmless water as a solvent. Xu et al. [20] used glycerol as a solvent, and cystine as a source of C, N, and S to prepare N, S–CD by microwave-assisted synthesis. A fluorescent N, S–CD probe for determination of 1–75 μM Hg(II) was proposed by using the aggregation-inducing enhancement effect of CDs, with an excitation/fluorescence wavelength of 325/385 nm and a detection limit of 0.5 μM. Li et al. [21] used ammonium citrate and L-cysteine to charge the current body in order to synthesize N, S–CD with blue fluorescence by microwave-assisted synthesis. Levofloxacin (LEV) can be detected by ratiometric fluorescence methods with a detection limit of 5.1 μg/L (3 σ/k) and a determination range of 0.01 to 70 mg/L. Yu et al. [22] used amino acids as raw materials, and controlled carbon and nitrogen composition and related chemical bonds, to synthesize carbon dots by microwave, and to determine 12.5–250 μM Fe^3+^. At present, the most important applications of carbon dots are clinical therapy, bioimaging, and fluorescence sensing [23,24,25]. Iannazzo et al. [24] reviewed graphene quantum dot synthesis and functionalization, and the application as nanoplatforms for anticancer therapy. Du et al. [25] introduced different synthetic methods for tuning the structure of doping carbon dots for applications in bioimaging. In fluorescence sensor analysis, most methods are based on the redox and complex reactions that result in CD fluorescence quenching or fluorescence enhancement. Yu et al. [26] invented a fluorescence resonance energy transfer (FRET) ratio fluorescence sensor for the detection of 1–10 mmol/L H_2_S. Ahmed et al. [27] reported thermal carbonization synthesis of carbon dots which is based on fluorescence quenching by 4-nitrophenol (4-NP) with a mixture of ethylene glycol bis-(2-aminoethyl ether)-N,N,N’,N’-tetraacetic acid (EGTA) and tris(hydroxymethyl) ethylenediamine to detect 0.1 to 50 μM 4-NP with a detection limit of 28 nM. Luo et al. [28] used a fluorescent chain-modified single-stranded nucleic acid and an Apt-modified graphene oxide to detect 10–800 nM ATP fluorescence. Cobalt oxyhydroxide (CoOOH) nanosheets are effective fluorescence quenchers due to their specific optical properties and specific surface areas, and were encapsulated with Apt-modified CD to detect 5–156 nM methyl propylamine (MTA) [29]. Shi et al. [30] used carbon dots as fluorescent labeling agents to modify complementary nucleic acids, and immobilized Apt on the surface of Fe_3_O_4_ nanoparticles to detect 0.25–50 ng/mL β-lactoglobulin with a detection limit of 37 pg/mL. However, there are no reports on a non-labeled Apt-mediated CD fluorescent probe or a catalytic 3,3′,5,5′-tetramethylbenzidine hydroxide oxidation product (TMB_OX_) probe for IPS.

3,3′,5,5′-Tetramethylbenzidine hydroxide (TMB) is a non-carcinogenic and non-mutagenic chromogenic agent [31]. At present, TMB mainly has been used for photometry in nanoanalysis. Lin et al. [32] developed a differential detection of 0.005–10 U/mL T4 polynucleotide kinase by a MnO_2_ nanosheet–TMB colorimetric system. Shi et al. [33] used carbon nanodots as catalysts to detect 0.002–0.10 mmol/L H_2_O_2_ and 0.0010–0.50 mM glucose by spectrophotometry. Ju et al. [34] designed a colorimetric sensor for 0.1-157.6 μM glutathione which is based on the peroxidase activity of silver nanoparticles on nitrogen-doped graphene quantum dots (AgNPs–N–GQDs). The reproductive toxicity, mutagenicity, carcinogenicity, cytotoxicity, genotoxicity, teratogenicity, and immunotoxicity of organophosphorus pesticides (OPPs) was investigated in all kinds of pesticides [35], and IPS was one of them. The detection techniques were mainly gas chromatography, high performance liquid chromatography, fluorescence, and electrochemical sensors [36,37,38,39]. Herein, on the basis of the Apt–IPS reaction, the CD fluorescence probe, and the TMB_OX_ probe, two new, rapid, and sensitive methods for the detection of IPS were established.

## 2. Materials and Methods

### 2.1. Apparatus

A model of Hitachi F-7000 fluorescence spectrophotometer (Hitachi High-Technologies Corporation, Tokyo, Japan) and a model of TU-1901 double beam UV-visible spectrophotometer (Beijing Purkingje General Instrument Co., Ltd., Beijing, China) were used to record the fluorescence and absorption spectra. The reaction was carried out in an HH-S2 electric hot water bath (Earth Automation Instrument Plant, Jintan, China). The characterization of nanoparticles was carried out on an S-4800 field emission scanning electron microscope (Hitachi High-Technologies Corporation, Japan/Oxford Company, Oxford, UK). The laser scattering was carried out on a Zeta Sizer Nano nanometer and particle size and zeta potential analyzer (Malvern Co., Malvern, UK). The sub-boiling water was obtained from a SYZ-550 quartz sub-boiling distiller (Crystal Glass Instrument Plant, Jiangsu, Nanjing, China). The carbon dots were synthesized on a WX-6000 microwave digestion instrument (Preekem Scientific Instruments Co., Ltd., Shanghai, China). 

### 2.2. Reagents

Nucleic acid aptamers (Apt) with the sequence: 5′-3′ AGC TTG CTG CAG CGA TTC TTG ATC GCC ACA GAG CT were purchased from Shanghai Sangon Biotech Co., Ltd. (Shanghai, China). Isocarbophos (IPS, 98.7% purity, NO: 20151113, GB(E)061673) was purchased from Beijing Century Aoke Biotechnology Co., Ltd. (Beijing, China). Profenofos, citric acid (AR), and urea (AR) were purchased from the National Pharmaceutical Group Chemical Reagents Company (Shanghai, China). Glyphosate was purchased from the Beijing Bailingwei Technology Co., Ltd. (Beijing, China). A 10 mmol/L AgNO_3_ solution, 0.1 mol/L sodium citrate solution, 30% H_2_O_2_ solution, 0.1 mol/L NaBH_4_ solution, and 3,3′,5,5′-tetramethylbenzidine (TMB, stored in 2–8 °C, T818493-5g, CAS: 54827-17-7) were purchased from Shanghai Maclean Biochemical Technology Co., Ltd. (Shanghai, China). A 0.2 mol/L pH 3.6 HAc–NaAc buffer solution, 0.2 mol/L NaH_2_PO_4_–Na_2_HPO_4_ buffer solution, 1.0 mol/L HCl solution, 0.25 mol/L NaOH solution, and 30 mg/L IPS standard solution were prepared. All reagents were of analytical reagent grade. All the solutions were prepared with ultrapure water.

### 2.3. Carbon Dot Preparation

CD_g_: Under ultrasonic irradiation, 1g glucose and 0.8 g urea were dissolved in 30 mL of water to form a transparent solution, which was then transferred to a digestion tank, sealed, and placed in a microwave digestion apparatus. The temperature was set at 140 °C, with a pressure of 4.5 atm, holding time of 10 min, and the irradiation time of 10 min. After completion of the operation, the mixture was cooled to room temperature to obtain a brownish yellow solution. It was dialyzed against a dialysis bag with a cut off molecular weight of 3500 Da for 12 h, and the C concentration, calculated as total amount of carbon, was 17 mg/mL CD_g_.

CD_N_: One gram of citric acid and 0.8 g of urea were sonicated in 30 mL of water to form a transparent solution, which was then transferred to a digestion tank, sealed, and placed in a microwave digestion apparatus at a temperature of 140 °C and a pressure of 4.5 atm, and irradiated for 10 min. Then, it was cooled to room temperature to obtain a pale yellow solution, for which the total amount of carbon was calculated to determine a concentration of 17 mg/mL CD_N_ solution.

CD_S_: One gram of trisodium citrate and 0.8 g of urea were sonicated in 30 mL of water to form a transparent solution, which was then transferred to a digestion tank, sealed, and placed in a microwave digestion apparatus at a temperature of 140 °C and a pressure of 5.0 atm. The holding time was 10 min and the irradiation time was 10 min. After completion of the reaction, it was cooled to room temperature to obtain a pale yellow clear solution, for which the total amount of carbon was calculated to determine a 13 mg/mL CD_S_ solution.

### 2.4. Procedure

CD probe: in a 5 mL test tube, 15 μg/L isocarbophos standard solution, 200 μL of 0.2 mol/L pH 7.4 NaH_2_PO_4_–Na_2_HPO_4_ buffer solution, 200 μL of 1.55 μmol/L Apt solution, and 100 μL of 0.1 mg/L carbon dot solution were combined and diluted to 1.5 mL with water. The fluorescence spectrum was measured at a specific excitation wavelength of each CD_S_, and the ΔF was calculated by the subtraction of blank F_0_ without isocarbophos (IPS) from F.

TMB_OX_ probe: in a 5 mL test tube, 1.5 μg/L IPS standard solution, 30 μL of a 1.55 μmol/L Apt, 100 μL of 1 mmol/L pH 3.6 HAc–NaAc buffer solution, 50 μL of 0.1 mg/L carbon dot solution, 40 μL of 2 mmol/L (0.006%) H_2_O_2_ solution, 50 μL of 0.5 mmol/L TMB solution, and 200 μL of 1 mmol/L pH 3.6 HAc–NaAc solutions were combined and diluted to 1.5 mL. The tube was heated in a 50 °C water bath for 15 min, the reaction stopped with ice water. Under the excitation wavelength of 285 nm, the voltage of 350 V, and the slit of 5 nm, the fluorescence spectrum was recorded. The fluorescence intensity at 406 nm was measured to be F_406 nm_. The blank (F_406 nm_)_0_ without analyte was recorded. ΔF_406 nm_ = F_406 nm_ − (F_406 nm_)_0_ was calculated. 

## 3. Results and Discussion

### 3.1. Analytical Principle

In the pH 3.6 HAc–NaAc buffer solution, the carbon dots had a strong catalytic effect on the reaction of H_2_O_2_/TMB to form TMB_OX_ oxidation products. When a certain concentration of the Apt was present, it adsorbed on the surface of the carbon dot, resulting in the CD catalytic action weakening. After the target molecule, IPS, was added it specifically bound to the Apt, and the CDs released to cause restoration of the catalytic action due to the affinity of Apt–IPS being larger than that of Apt–CD. With the increase of the concentration of IPS, the higher the desorption of CDs, the faster the catalytic reaction of H_2_O_2_/TMB, the greater the concentration of TMB_OX_ formed, and the fluorescent signal gradually increased. Using TMB_OX_ as a fluorescent probe and the catalytic effect of CD to amplify the signal, a new and highly sensitive fluorescence method for determination of IPS was established (Figure 1). Using CD_N_ as a fluorescent probe, based on the fluorescence enhancement of Apt–IPS–CD_N_ reaction in the pH 7.4 NaH_2_PO_4_–Na_2_HPO_4_ buffer solution, a new and simple label-free Apt CD fluorescence method for the rapid determination of IPS was also established.

### 3.2. Fluorescence Spectra

The fluorescence properties of CD_g_, CD_N_ and CD_S_ were examined. For CD_g_, 355 nm was used as λ_ex_, and with a voltage of 400 V and a slit of 10 nm, a fluorescence peak was generated at 457.2 nm. As the concentration increased, the intensity of the fluorescence peak gradually increased (Appendix A). With λ_ex_ of 350 nm, voltage of 350 V, and a slit of 10 nm, CD_N_ produced a fluorescence peak at 440 nm. As the concentration increased, the intensity of the fluorescence emission peak gradually increased (Figure 2). Under λ_ex_ of 370 nm, voltage of 400 V, and a slit of 5 nm, CD_S_ generated a fluorescence peak at 440 nm (Appendix A). According to the slope of the regression equation (Table 1), due to its molecular weight uncertainty, the fluorescence of CD_N_ was strongest, followed by CD_S_, and the dynamic range of CD_S_ was narrower. Since N belongs to an electron donating atom, the electron cloud density around the nitrogen atom in the nitrogen-doped carbon dots made it have good electron conductivity. Under the excitation of ultraviolet light, more electrons in CD_N_ transitioned from the ground state to the excited state. Due to the excited state being unstable, the electrons released energy in the form of fluorescence, returning to the ground state. The fluorescence intensity of CD_N_ was significantly enhanced compared with the fluorescence intensity of non-N-doped CD.

The Apt–IPS–CD_N_ system exhibited a fluorescence peak at 440 nm ascribed to CD_N_, with λ_ex_ of 350 nm, a voltage of 350 V, and a slit of 10 nm. As the concentration of IPS increased, the more carbon dots were released, and the stronger the fluorescent signal (Figure 3). The fluorescence wavelength was selected to detect IPS. When λ_em_ was 440 nm, there was an excitation peak at 350 nm.

In the catalytic system, in addition to carbon dot fluorescence, TMB_OX_ also has strong fluorescence. However, the carbon dot concentration was very low and the fluorescence signal was negligible. When the CD_N_ concentration was high, such as 50 mg/L, there was no catalytic effect on H_2_O_2_–TMB. The Apt–IPS–CD_N_–H_2_O_2_–TMB catalytic analytical system generated a fluorescence peak at 400 nm, at an excitation wavelength of 285 nm, a voltage of 350 V, and a slit of 5 nm. As the IPS concentration increased, the fluorescence emission peak intensity gradually increased. There are similar correlations in CD_g_ and CD_S_ systems (Figure 4A, Appendix A). In the CD_N_–H_2_O_2_–TMB catalytic system, an excitation peak was generated at 350 nm with a 500 nm emission wavelength, a voltage of 350 V, and a slit of 10 nm. As the concentration of CD increased, the intensity gradually increased (Figure 4B, Appendix A). When a suitable concentration of Apt was added, Apt encapsulated the carbon dots to inhibit the catalytic ability of the carbon dots, resulting in fluorescence intensity decreasing due to TMB_OX_ decreasing (Figure 4C, Appendix A).

### 3.3. Nanocatalysis and Aptamer Inhibition

Citric acid has –COOH and –OH polar groups and urea contains –C=O and –NH_2_. Since both citric acid and urea are polar molecules, when the microwave energy acts on the molecules to generate heating, O=C–NH bonds are produced, because citric acid has many –COOH bonds and urea has a plurality of –NH_2_, and the dehydration reaction easily occurs with the microwave heating process. Therefore, the different monomers will soon be polymerized by dehydration between the monomers to form a certain degree of polymerization of the amide species. As the microwave energy continues to accumulate, the amides of different polymerization degrees will undergo a significant carbonization process to form nitrogen-doped carbon dots.

Since N is an electron donating atom, the electron cloud density around the nitrogen atom in the nitrogen-doped carbon dots caused it to have good electron conductivity, and it also exhibited unique properties in catalytic reactions. Under certain conditions, H_2_O_2_ and TMB had difficulty reacting. When nanoparticles such as carbon dots were added, the surface electrons of the CD_N_ played a catalytic role that enhanced the redox-electron transfer. As the concentration of the catalyst increased, the catalytic ability increased, and the fluorescence gradually increased due to TMB_OX_ increase. When the Apt was added, more carbon dots were entrapped, resulting in inhibition of catalysis (Table 2). The catalytic mechanism is shown in Figure 5.

### 3.4. Scanning Electron Microscopy, Transmission Electron Microscopy, Laser Scattering and Infrared Spectroscopy

The prepared carbon dots were diluted and dripped onto the silicon wafer for electron microscopy scanning. Because the conductivity was very poor, the gold spray treatment with an average particle size of 20 nm was added to conduct SEM of CD_N_ (Figure 6A). Transmission electron microscopy indicated that the average particle size was 25 nm (Figure 6B). The laser scattering graph showed that the CD size distribution was from 10 nm to 30 nm, with an average particle size of 27 nm (Figure 6C). The carbon dots were prepared according to the experimental method, and were placed in a material tray, pre-frozen in a cold trap for 5 h by the vacuum drying freezer, then dried at 0.1 Pa for 24 h, and the obtained solid sample was mixed with a certain amount of KBr, then ground in an agate mortar for 2 to 5 min, and the powder was tableted by a tableting machine. The tablet was removed by a blade and loaded into a tablet holder, and the spectrum was recorded by an infrared spectrometer with a KBr blank sheet as a reference. It could be seen from the infrared spectrum (Figure 6D) that CD_N_ had absorption at 3030 cm^−1^, which was ascribed to the stretching vibration of N–H, O–H, and C–H on unsaturated carbon. The absorption peak of 3396 cm^−1^ was a symmetric stretching vibration of N–H. The absorption peaks of 1385 cm^−1^ and 627 cm^−1^ may have been the bending vibration of O–H. The absorption peak at 1573 cm^−1^ indicated the presence of a C=C conjugated structure. The absorption peak of 1111 cm^−1^ may have been the stretching vibration of C–O.

### 3.5. Optimization of the Analytical Conditions

For the H_2_O_2_–TMB catalytic system, the effect of Apt, CD_N_, H_2_O_2_, and TMB concentrations, pH and its buffer solution concentration, reaction temperature, and time on the fluorescence signal was investigated. Results (Appendix A) showed that a 0.031 μmol/L Apt, 3.33 μg/L CD_N_, 0.053 mmol/L H_2_O_2_, 0.017 mmol/L TMB, 0.13 mmol/L pH 3.6 HAc–NaAc buffer solution, and a reaction temperature of 50 °C for 15 min gave the largest ΔF_406 nm_, and were chosen for use (Table 3).

For the Apt–IPS–CD_N_ system, the analytical conditions were examined (Appendix A). When HAc–NaAc was used as a buffer solution, the fluorescence signal does not change much in the pH range of 4.8–6.0. When NaH_2_PO_4_–Na_2_HPO_4_ was used as a buffer solution, results showed that the fluorescence signal was maximal for pH 7.4 NaH_2_PO_4_–Na_2_HPO_4_ buffer solution, and so it was selected for use. The effect of the amount of pH 7.4 NaH_2_PO_4_–Na_2_HPO_4_ buffer solution on the fluorescence signal of the system was investigated. A 0.027 mol/L buffer solution was selected. CD_N_ was a probe in the system. When CD_N_ concentration was 7.3 mg/L, the fluorescence signal was the strongest, so 7.3 mg/L CD_N_ was used (Table 3).

### 3.6. Working Curve

Under optimal conditions, the relationship between the IPS concentration and its corresponding ΔF was obtained (Table 4). In the three catalytic systems, the slope of the ΔF working curve of CD_N_ system was the largest. Therefore, the CD_N_ system was the most sensitive and could be used for fluorescence detection of 0.025–1.5 μg/L IPS, with a detection limit (DL) of 0.015 μg/L IPS, which was defined as 3 times the standard deviation (3 σ) that was obtained from 5 replicates of the parallel blank. So the catalytic fluorescence system was selected to determine the concentration of IPS in the sample. This method was simpler and more sensitive than the reported spectral method for determining IPS (Table 5). Among the systems of Apt–IPS–CD_g_/CD_N_/CD_S_, the slope of the ΔF working curve of the Apt–IPS–CD_N_ system was the largest, so the CD_N_ system was the most sensitive and could be used for fluorescence detection of 0.25–1.5 μg/L IPS, with a detection limit of 0.11 μg/L IPS. In addition, the microwave preparation of CD_N_ was simple and rapid (Table 6).

### 3.7. Influence of Interfering Ions

For the H_2_O_2_–TMB nanocatalytic analytical fluorescence system, the interference of coexisting ions on the fluorescence measurement of 0.1 μg/L IPS was investigated experimentally. Results showed that concentrations of 1000 times Zn^2+^, Ca^2+^, Ni^2+^, Co^2+^, Ba^2+^, Mg^2+^, K^+^, glyphosate, tributylphosphine, and profenofos, 500 times Mn^2+^, CO_3_^2−^, HCO_3_^−^, NO_2_^−^, and Al^3+^, 250 times NH_4_Cl, Fe^3+^, Bi^3+^, Cu^2+^, and Pb^2+^, and 100 times Cr^6+^, Fe^2+^, and Hg^2+^ had no interference on the determination of IPS (Appendix A). For the Apt–IPS–CD_N_ fluorescence system, influences of coexisting ions on the fluorescence measurement of 0.5 μg/L IPS were also examined experimentally. Results showed that concentrations of 1000 times Zn^2+^, Ca^2+^, Ni^2+^, Ba^2+^, Mg^2+^, K^+^, and glyphosate, 500 times Mn^2+^, NH_4_Cl, CO_3_^2−^, HCO_3_^−^, NO_2_^−^, tributylphosphine, and profenofos had no interference on the determination of IPS (Appendix A). The specificity of the nanocatalytic analytical fluorescence system was better than the CD_N_ system, especially for other organic phosphoruses such as tributylphosphine and profenofos.

### 3.8. Sample Analysis

Domestic sewage, farmland water, and pond water were collected in 50 mL samples, filtered to remove suspended particles, and stored at 5 °C. The original pH of the sample was 5.6 and was measured by pH meter, and was adjusted to pH 7 with 0.1 mol/L NaOH solution. Using the Apt–IPS–CD_N_–H_2_O_2_–TMB fluorescence method, the water samples were analyzed five times in parallel. The results (Appendix A) show that the recovery was 98.5–104%, and the relative standard deviation was 2.4–2.9%.

## 4. Conclusions

In this paper, CD_N_ with high fluorescence were synthesized by the microwave digestion procedure. Based on the catalytic action of CD_N_ on the H_2_O_2_/TMB fluorescence reaction and the specific binding of IPS to Apt, two new fluoresce methods were established for analysis of IPS at the μg/L level. These methods have the advantages of simple operation, high sensitivity, and good selectivity.

## Figures and Tables

**Figure 1 nanomaterials-09-00223-f001:**
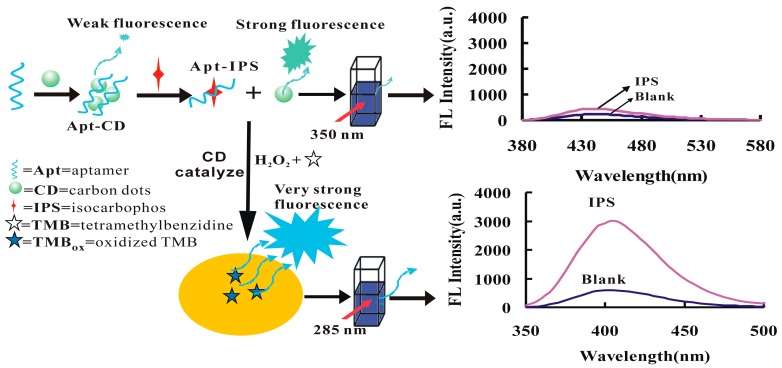
Principle of carbon dot (CD) and 3,3′,5,5′-tetramethylbenzidine hydroxide oxidation product (TMB_OX_) probes for isocarbophos (IPS) based on the aptamer (Apt) reaction.

**Figure 2 nanomaterials-09-00223-f002:**
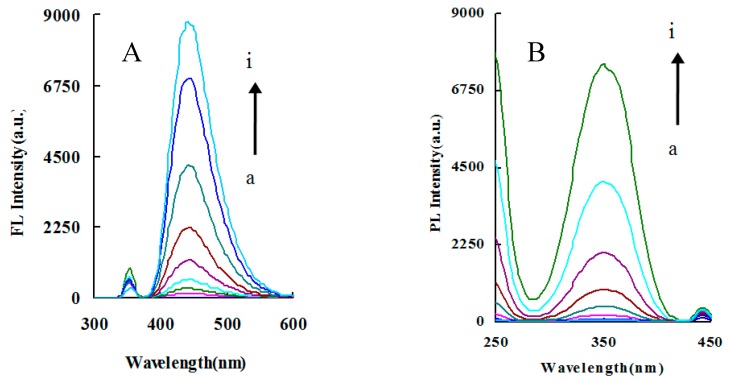
Fluorescence (**A**) and excited (**B**) spectra of CD_N_. (a) 0 mg/L CD_N_; (b) 129.2 mg/L CD_N_; (c) 265.2 mg/L CD_N_; (d) 530.4 mg/L CD_N_; (e) 1060.8 mg/L CD_N_; (f) 2128.4 mg/L CD_N_; (g) 4250 mg/L CD_N_; (h) 8500 mg/L CD_N_; (i) 17,000 mg/L CD_N_.

**Figure 3 nanomaterials-09-00223-f003:**
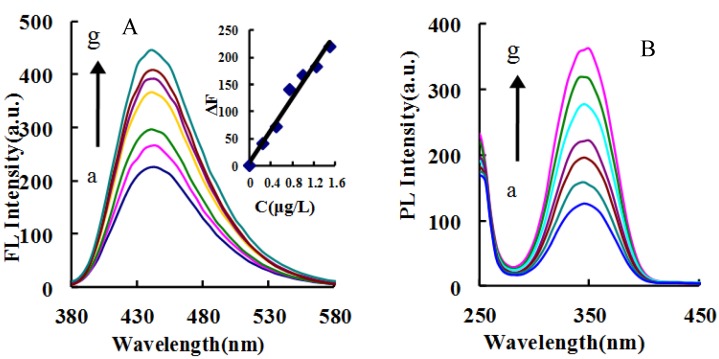
Fluorescence (**A**) and excited (**B**) spectra of the Apt–IPS–CD_N_ system. (a) 0.21 μmol/L Apt + 11.28 mg/L CD_N_ + 0.027 mol/L NaH_2_PO_4_–Na_2_HPO_4_; (b) a + 0.25 μg/L IPS; (c) a + 0.5μg/L IPS; (d) a + 0.75μg/L IPS; (e) a + 1.0 μg/L IPS; (f) a + 1.25 μg/L IPS; (g) a + 1.5μg/L IPS.

**Figure 4 nanomaterials-09-00223-f004:**
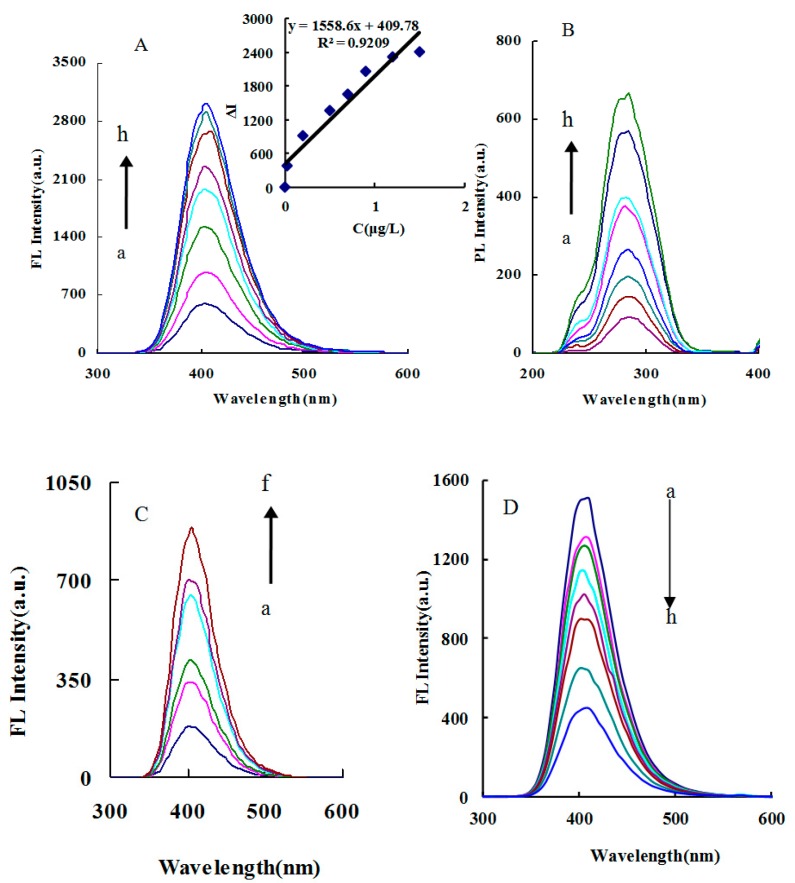
Fluorescence of the CD_N_ catalytic system (**A**) Fluorescence spectra of the Apt–IPS–CD_N_–H_2_O_2_–TMB catalytic analytical system, a: 31 nmol/L Apt + 0.45 mg/L CD_N_ + 0.053 mmol/L H_2_O_2_ + 0.017 mmol/L TMB + 0.13 mmol/L pH 3.6 HAc–NaAc; b: a + 0.025 μg/L IPS; c: a + 0.1μg/L IPS; d: a + 0.3μg/L IPS; e: a + 0.5μg/L IPS; f: a + 0.7μg/L IPS; g: a + 0.9μg/L IPS; h: a + 1.2μg/L IPS. (**B**) Excited spectra of A. (**C**) Fluorescence spectra of the CD_N_–H_2_O_2_–TMB catalytic system, a: 0.13 mmol/L H_2_O_2_+33 μmol/L TMB + 0.13 mmol/L pH 3.6 HAc–NaAc; b: a + 0.028 mg/L CD_N_; c: a + 0.057 mg/L CD_N_; d: a + 0.113 mg/L CD_N_; e: a + 0.17 mg/L CD_N_; f: a + 0.34 mg/L CD_N_. (**D**) Fluorescence spectra of the Apt–CD_N_–H_2_O_2_–TMB system, a: 0.34 mg/L CD_N_ + 0.053 mmol/L H_2_O_2_ + 0.017 mmol/L TMB + 0.13 mmol/L pH 3.6 HAc–NaAc; b: a + 5.17 nmol/L Apt; c: a + 7.23 nmol/L Apt; d: a + 10.33 nmol/L Apt; e: a + 15.5 nmol/L Apt IPS; f: a + 20.67 nmol/L Apt; g: a + 25.83 nmol/L Apt; h: a + 31 nmol/L Apt.

**Figure 5 nanomaterials-09-00223-f005:**
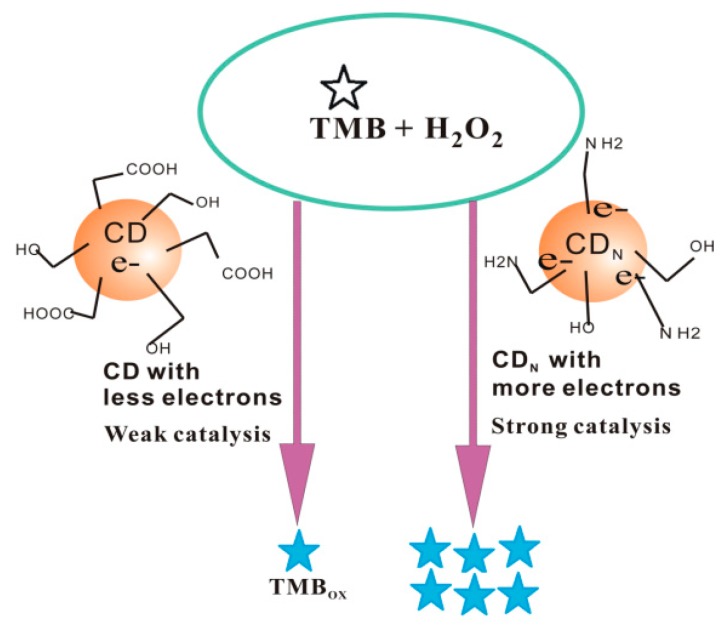
Mechanism of catalytic reaction of nitrogen-doped carbon dots.

**Figure 6 nanomaterials-09-00223-f006:**
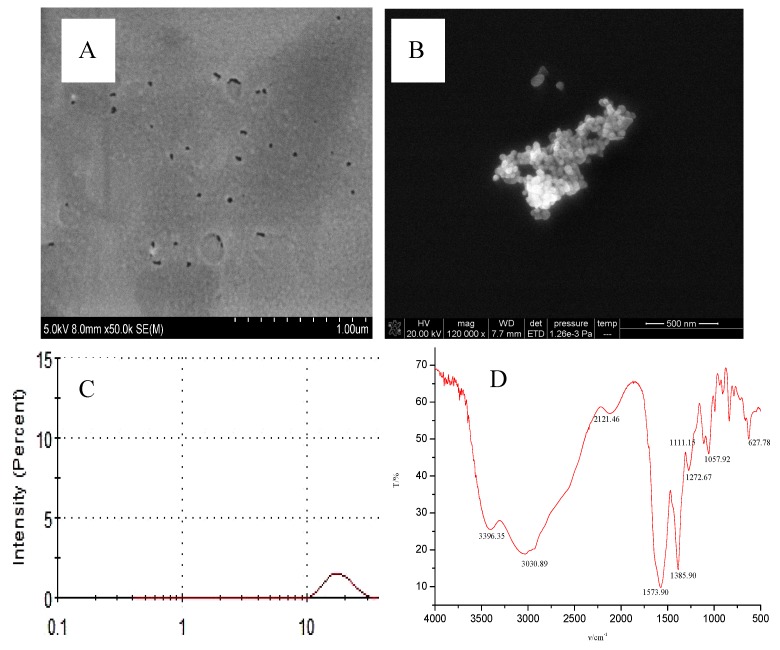
(**A**) SEM, (**B**) TEM, (**C**) laser scattering, and (**D**) IR of CD_N_.

**Table 1 nanomaterials-09-00223-t001:** Comparison of CD fluorescence characteristics.

CD	Determination Range (mg/L)	Regression Equation	Coefficient
CD_N_	6.24–3250	ΔF_440 nm_ = 0.65C + 190.7	0.8639
CD_S_	14.1–353.6	ΔF_457.2 nm_ = 0.61C + 28.8	0.9058
CD_g_	129.2–17,000	ΔF_440 nm_ = 0.546C + 643.9	0.9171

**Table 2 nanomaterials-09-00223-t002:** Comparison of nanocatalysis and aptamer inhibition characteristics.

Nanocatalytic System	Dynamic Range (µg/L)	Regression Equation
CD_g_	23–227	ΔF_406 nm_ = 1406.8C_CD_ + 69.5
CD_N_	28–340	ΔF_406 nm_ = 3167.4C_CD_ + 40.3
CD_S_	9–347	ΔF_406 nm_ = 1531.7C_CD_ + 62.9
Apt-CD_g_	5.17–25.83	ΔF_406 nm_ = 30.4C_Apt_ − 48. 2
Apt-CD_N_	5.17–31	ΔF_406 nm_ = 32.7C_Apt_ − 1.6
Apt-CD_S_	5.17–31	ΔF_406 nm_ = 24.2C_Apt_ − 54.8

**Table 3 nanomaterials-09-00223-t003:** Optimization of the analytical conditions.

System	Parameters	Range	Best Value
IPS–Apt–CD_N_–H_2_O_2_–TMB	Apt concentration	0–0.052 μmol/L	0.031 μmol/L
	CD_N_ concentration	0–10 µg/L	3.33 μg/L
	H_2_O_2_ concentration	0–0.16 mmol/L	0.053 mmol/L
	TMB concentration	0–0.05 μmol/L	0.017 mmol/L
	pH	3.2–5.8	3.6
	HAc–NaAc buffer solution	0–0.67 mmol/L	0.13 mmol/L
	Temperature	20–80 °C	50 °C
	Reaction time	5–30 min	15 min
IPS–Apt–CD_N_	pH	3.2–8	7.4
	NaH_2_PO_4_–Na_2_HPO_4_ buffer solution	0–0.04 mol/L	0.027 mol/L
	Apt concentration	0–0.52 μmol/L	0.21 μmol/L Apt
	CD_N_ concentration	0–28 mg/L	7.3 mg/L CD_N_
	NaH_2_PO_4_–Na_2_HPO_4_ buffer solution		0.027 mol/L

**Table 4 nanomaterials-09-00223-t004:** Comparison of analytical characteristics for the IPS methods.

System	Determination Range (μg/L)	Regression Equation	Coefficient	DL (μg/L)
Apt–CD_g_–H_2_O_2_–TMB	0.1–1.1	ΔF_406 nm_ = 873.5C_IPS_ + 20.0	0.9873	0.04
Apt–CD_N_–H_2_O_2_–TMB	0.025–1.5	ΔF_406 nm_ = 1558.6C_IPS_ + 40.9	0.9209	0.015
Apt–CD_S_–H_2_O_2_–TMB	0.12–2	ΔF_406 nm_ = 603.4C_IPS_ + 88.2	0.8928	0.039
Apt–CD_N_	0.25–1.5	ΔF_440 nm_ = 148.0C_IPS_ + 6.1	0.9759	0.11
Apt–CD_g_	0.5–3.0	ΔF_435 nm_ = 2.25C_IPS_ + 0.4	0.9549	0.23
Apt–CD_S_	0.5–3.0	ΔF_440 nm_ = 31.2C_IPS_ + 7.1	0.9243	0.21

**Table 5 nanomaterials-09-00223-t005:** Comparison of molecular spectral methods for determination of IPS.

Method	Principle	LR (μg/L)	DL (μg/L)	Annotation	Ref.
Fluorescence analysis	Based on the fluorescence quenching of CdSe quantum dots detection of IPS.	67–3153	31.8	High precision, but low sensitivity.	[40]
Fluorescence analysis	Apt recognized IPS is fluorescently labeled, and when it binds to a quencher group on the complementary DNA strand, the fluorescent is attenuated, and when the Apt recognizes and binds the target, the fluorescent is recovered.	1.4 × 10^4^–1.44 × 10^5^	0.33 × 10^4^	Fast, simple, low sensitivity.	[41]
Chemiluminescence method	Organophosphorus insecticide sample was injected into a column using methanol/water eluent, based on the chemiluminescence reaction of IPS–luminol–H_2_O_2_.	86–1.5 × 10^4^	50	High sensitivity, but complicated operation.	[42]
SERS	Apt was modified nanosilver, and 6-mercaptoethanol (MH) was backfilled to prevent non-specific binding, resulting in the SERS effect, and amphetamine combination with Apt. MH moved away from the silver surface, causing the SERS to decrease.	—	982.6	Fast, selective, but not very sensitive.	[43]
TMB_OX_ probe	Apt was used to modulate the CD_N_ catalysis to generate the TMB_OX_ fluorescent probe to detect IPS.	0.025–1.5	0.015	High sensitivity, good selectivity.	This method
CD probe	Used Apt to adjust CD fluorescence to detect IPS.	0.25–1.5	0.11	Sensitive, selective, and simple.	This method

**Table 6 nanomaterials-09-00223-t006:** Comparison of preparation procedures for CD_N_.

Procedure	C Source	N Source	Time	Ref
Hydrothermal	3-(3,4-dihydroxyphenyl)-l-alanine	3-(3,4-dihydroxyphenyl)-l-alanine	300 °C for 2 h	[44]
Carbonization	CCl_4_	1,2-ethylenediamine	200 °C for 2 h	[45]
Microplasma	Citric acid	Ethylenediamine	60 min with argon	[46]
Ultrasonic	Glucose	Aqua ammonia	24 h at room temperature	[47]
Microwave	Citric acid	Ethylenediamine	140 °C for 15 min.	[48]
Microwave	Citric acid	Urea	140 °C for 10 min.	This procedure

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
