# Peer review of "Using N-doped Carbon Dots Prepared Rapidly by Microwave Digestion as Nanoprobes and Nanocatalysts for Fluorescence Determination of Ultratrace Isocarbophos with Label-Free Aptamers"

_nanomaterials, 2019, doi:10.3390/nano9020223_

Reviewer 1 Report

This is an interesting work devoted to method development. Both methods proposed, i.e. Apt-CD(N)-H2O2-TMB and Apt-CD(N), provided relatively low DLs for IPS as compared to other methods reported in literature. Th emethod of synthesis of CD is also uncomplicated. The method is in general properly validated and its analytical application is shown on the example of domestic sewage. There are only few comments to be addressed:

1. p. 8, l. 233. “When adding different pH” sounds strange. The text of the article should be subjected to the proofreading.

2. p. 7, l. 220 and p. 8, l. 237. It is not suitable for the robust analytical method that such a strict pH should be provided.

3. p. 8, l. 247. What was the reason for such narrow linearity ranges for both selected methods, i.e. Apt-CD(N)-H2O2-TMB and Apt-CD(N)?

4. Table 3 and the related text. How was the DL defined and assessed?

5. p. 10, l. 259. What about specificity of Apt-CD(N)-H2O2-TMB and Apt-CD(N) methods?

6. p. 10, l. 271. What was original pH of the sample taken for analysis? How was it verified that indeed pH of the prepared sample solution was 3.6?

7. p. 11, l. 308, 313, 319. The concentration of C was calculated assuming that processes involved were 100% efficient. Why was not this concentration determined to assess the actual concentration of CD?

Author Response

1. p. 8, l. 233. “When adding different pH” sounds strange. The text of the article should be subjected to the proofreading.

Answer: “When adding different pH of NaH2PO4-Na2HPO4 buffer solution 200 μL, 1.5 μg/L IPS, 0.21 μmol/L Apt, 7.3 mg/L CDN, to a volume of 1.5 mL, according to CDN fluorescence characteristics, selected λex = 350 nm, voltage at 350 V and a slit of 10 nm, the fluorescence peak at 440 nm was observed. The fluorescence signal changes when different pH buffers are added, so a pH 7.4 NaH2PO4-Na2HPO4buffer solution was selected.” was changed to “When NaH2PO4-Na2HPO4 was used as buffer solution, results show that the fluorescence signal was maximal for pH 7.4 NaH2PO4-Na2HPO4 buffer solution, and was selected for use.” in red letters. And the text was revised carefully.

2. p. 7, l. 220 and p. 8, l. 237. It is not suitable for the robust analytical method that such a strict pH should be provided.

Answer: The pH tests were repeated and the Figures were revised as in Figure S13, S14, S18 and S19.

3. p. 8, l. 247. What was the reason for such narrow linearity ranges for both selected methods, i.e. Apt-CD(N)-H2O2-TMB and Apt-CD(N)?

Answer: The linearity range is related to the fluorescence property of TMBOX and CDN probes, and the Apt-CD(N)-H2O2-TMB and Apt-CD(N) systems have a big slope for the working curves. The linearity range of Apt-CD(N)-H2O2-TMB system with 60 times (=1.5/0.025) is wider than the Apt-CD(N) system, the detection limit is low, and was selected to determine the IPS in the sample.

4. Table 3 and the related text. How was the DL defined and assessed?

Answer: DL is the abbreviation of detection limit, and it was defined as 3 times of standard deviation that was obtained from 5 times of the parallel blank. That was added in the section of 3.6.

5. p. 10, l. 259. What about specificity of Apt-CD(N)-H2O2-TMB and Apt-CD(N) methods?

Answer: The specificity of nanocatalytic analytical fluorescence system was better than the CDN system, especially for other organic phosphorus such as tributylphosphine and profenofos. This was added in the section of 3.7.

6. p. 10, l. 271. What was original pH of the sample taken for analysis? How was it verified that indeed pH of the prepared sample solution was 3.6?

Answer: The pH of the analytical system was measured by pH meter, the pH of the taken sample was 5-6 that was adjusted to pH 7 with 0.1 mol/L NaOH solution. This was added in the section of 3.8.

7. p. 11, l. 308, 313, 319. The concentration of C was calculated assuming that processes involved were 100% efficient. Why was not this concentration determined to assess the actual concentration of CD?

Answer: CD is complicated organic mixture, its structure, composition and molecular weight are uncertain, in general the concentration was calculated as total carbon.

Reviewer 2 Report

A label-free apatassay for quantitation of the pesticide isocarbophos (IPS) by using the fluorescent properties of N-doped carbon dots is reported. The oligonucleotide receptor is initially adsorbed onto the surface of carbon dots reducing their natural fluorescence. In the presence of IPS, the aptamer binds the IPS and fluoresce of CD is restored. Furthermore, a second approach based on the catalytic effect of these CDs in the reaction between H2O2 and TMB resulting in fluorescent TMBox is evaluated.

Important issues to be addressed are:

(1) In both schemes, the displacement of the aptamer from the carbon dot surface strongly depends on the affinity of the complex aptamer-IPS, which is the affinity constant of these complex in solution?

(2) The assay selectivity should be thoroughly studied. Could a temperature of 50ºC (the best value according to authors’ studies) force the aptamer displacement from the CD surface even in the absence of the analyte? Likewise, the effect of DNA fragments different from that of the aptamer should be tested in order to check the assay selectivity.

(3) Unlike authors state in the abstract, changes in fluorescence are by no means linear. Please, check and reassess all your results, the “linear” ranges are wrong.

(4) In section 2.5, the optimization of many different parameters is described; however, descriptions are very repetitive and incomplete. For the sake of clarity and fluency, it could be substituted by a table indicating the different parameters evaluated, the values tested (ranges), as well as the optimal values founded.

(5) It should be indicated at what levels it is necessary to determine the pesticide isocarbophos (IPS)?

(6) For fluorescence measurements, sometimes the slit is fixed to 10 nm, while in other cases the slit is 5 nm, what is the reason for these changes?

(7) Abbreviation CDg appears for the first time in page 3 without explaining the meaning.

Author Response

(1) In both schemes, the displacement of the aptamer from the carbon dot surface strongly depends on the affinity of the complex aptamer-IPS, which is the affinity constant of these complex in solution?

Answer: CD is complicated organic mixture, and its accurate concentration is difficult to obtain. Thus, the affinity constant of these complex of aptamer- CD and aptamer-IPS is difficult to obtain. Based on our results, the complex of aptamer-IPS is more stable than the aptamer- CD, and the affinity constant of aptamer-IPS is bigger than the aptamer-CD. This was added in the section of 3.1.

(2) The assay selectivity should be thoroughly studied. Could a temperature of 50ºC (the best value according to authors’ studies) force the aptamer displacement from the CD surface even in the absence of the analyte? Likewise, the effect of DNA fragments different from that of the aptamer should be tested in order to check the assay selectivity.

Answer: Thanks! Unlike protein, nucleic acid aptamer is stable, especially for temperature. From Figure 4B, we can see that the blank is small due to the aptamer tight coupling with CD with very weak catalysis at the temperature. The 300 and 100 times DNA fragment (CAT CTC TTC TCC GAG CCG) had no interference on the determinations, and was added in Table S1 and S2 respectively.

(3) Unlike authors state in the abstract, changes in fluorescence are by no means linear. Please, check and reassess all your results, the “linear” ranges are wrong.

Answer: Thanks! Changes in fluorescence are by no means linear. In fluorescence quantitative analysis method, linear working curve is necessary, because the relationship between fluorescence intensity and analyte concentration is simple and calculated conveniently. The “linear ranges” for assay of IPS was changed to “determination ranges” in the content.

(4) In section 2.5, the optimization of many different parameters is described; however, descriptions are very repetitive and incomplete. For the sake of clarity and fluency, it could be substituted by a table indicating the different parameters evaluated, the values tested (ranges), as well as the optimal values founded.

Answer: The section was revised, and Table 3 was added in section of 3.5.

(5) It should be indicated at what levels it is necessary to determine the pesticide isocarbophos (IPS)?

Answer: In the section of 4, IPS content at μg/L level can be determined.

(6) For fluorescence measurements, sometimes the slit is fixed to 10 nm, while in other cases the slit is 5 nm, what is the reason for these changes?

Answer: Because the maximum value of the fluorescence intensity of the fluorometer used in the experiment is 10000, sometimes when the slit is 10 nm, the fluorescence intensity is too big to record, and the slit can only be adjusted to 5 nm to obtain fluorescence data.

(7) Abbreviation CDg appears for the first time in page 3 without explaining the meaning.

Answer: After adjusted Experimental section before the result section, the meaning of CDg, CDN and CDs was clear in the section 2.3.

Reviewer 3 Report

The authors report the synthesis of N-doped carbon dots by microwave irradiation and their catalytic activity in the fluorescence reaction of tetramethylbenzidine hydroxide. The issues treated in the paper have relevance in biosensors field. I believe that the paper could be accepted for publication in this Journal after addressing some issues.

Firstly, the introduction section must be improved by adding more recent and relevant literature data on the use of carbon dots in biomedical field. As example, I would suggest the authors to consider the following reviews: a) J. Zhang, Carbon dots: large-scale synthesis, sensing and bioimaging, Materials Today, 2016, 19, 382; b) D. Iannazzo et al., Graphene quantum dots: multifunctional nanoplatforms for anticancer therapy, J. Mater. Chem. B, 2017, 5, 6471; c) Y. Du, Chemically doped fluorescent carbon and graphene quantum dots for bioimaging, sensor, catalytic and photoelectronic applications, Nanoscale, 2016, 5, 2532.

The major concerns regard the characterization of carbon dots. In particular, the authors assert that their average particle size is of approximately 20 nm and to support this data, they report only a SEM image with a scale bar in the order of micrometer. More detailed morphological investigations such TEM or AFM should be given in order to prove the reported particle size as well as DLS measurements. Moreover, the authors assert that the FTIR spectra in Figure 6 shows the presence of the carboxyl group because of the IR absorption at 3030 cm−1. How the authors can be sure of the presence of a carboxyl group when it is absent the representative peak of the carbonyl (C=O) bond in the range 1900-1600 cm-1 ?

Finally, the manuscript requires a careful revision. As example, the acronym of pesticide IPS must be defined and the legend in Figure 1 must be better be detailed since it is not clear the difference between the two fluorescence graphs.

Author Response

Firstly, the introduction section must be improved by adding more recent and relevant literature data on the use of carbon dots in biomedical field. As example, I would suggest the authors to consider the following reviews: a) J. Zhang, Carbon dots: large-scale synthesis, sensing and bioimaging, Materials Today, 2016, 19, 382; b) D. Iannazzo et al., Graphene quantum dots: multifunctional nanoplatforms for anticancer therapy, J. Mater. Chem. B20175, 6471; c) Y. Du, Chemically doped fluorescent carbon and graphene quantum dots for bioimaging, sensor, catalytic and photoelectronic applications, Nanoscale20165, 2532.

Answer: Thank you! I have added it [12, 24 and 25] in the introduction in red letters.

The major concerns regard the characterization of carbon dots. In particular, the authors assert that their average particle size is of approximately 20 nm and to support this data, they report only a SEM image with a scale bar in the order of micrometer. More detailed morphological investigations such TEM or AFM should be given in order to prove the reported particle size as well as DLS measurements. Moreover, the authors assert that the FTIR spectra in Figure 6 shows the presence of the carboxyl group because of the IR absorption at 3030 cm−1. How the authors can be sure of the presence of a carboxyl group when it is absent the representative peak of the carbonyl (C=O) bond in the range 1900-1600 cm-1 ?

Answer: The transmission electron microscopy and laser scattering were carried out.The transmission electron microscopy in indicated that the average particle size is 25 nm (Figure 6B). The laser scattering graph show that the CD size distribution is from 10nm to 30 nm, with an average particle size of 27 nm (Figure 6C).” was added in the section of 3.4. The “indicating the presence of a carboxyl group” was changed to “it is ascribing to stretching vibration of N-H, O-H and C-H on unsaturated carbon”.

Finally, the manuscript requires a careful revision. As example, the acronym of pesticide IPS must be defined and the legend in Figure 1 must be better be detailed since it is not clear the difference between the two fluorescence graphs.

Answer: The acronym of pesticide IPS was defined in the Abstract. Figure 1 was revised to clear the difference between the two fluorescence graphs with same ordinate scale etc.

Besides, we have adjusted Experimental section before the result section.

Round  2

Reviewer 2 Report

There are still some parts of the manuscript where authors describe a linear variation of the fluorescence with the concentration; however, a regression coefficient (R) lower than 0.995 is indicated. Please, revise your data, adjustments for table 2. Likewise, in figure S4 it seems that the signal is levelled off at 1 micrograms/L; therefore, no linear response up to 2 micrograms/L must be depicted, the dynamic range is narrower than described.

Author Response

Answer: ThanksThe “linear” was revised in the text in red letters. In Table 2, the “linear range” was changed to “dynamic range” and the column of “Coefficient” was deleted. The inserted figure was deleted from FigureS4.

Reviewer 3 Report

I believe that this revised version of the manuscript can be accepted for publication in this Journal.

Author Response

Answer: Thanks